Endorsement of reporting guidelines and clinical trial registration across urological medical journals: a cross-sectional study

Hagood Alex 1 alexhagoodresearch@gmail.com
Case Joseph 1
Magee Trevor 1
Smith Caleb 1
Nees Danya 1
Hughes Griffin 1
Vassar Matt 1 2
1 Office of Medical Student Research, Oklahoma State University College of Osteopathic Medicine , Tulsa , United States
2 Department of Psychiatry and Behavioral Sciences, Oklahoma State University College of Osteopathic Medicine , Tulsa, Oklahoma , United States
Gray Andrew
Electronic publication date: 2024 Dec 10
Publication date: 2024
Volume: 12
Electronic Location ID: e18619
Received 2024 Jan 10; Accepted 2024 Nov 11
Copyright: © 2024 Hagood et al.
Copyright year: 2024
Copyright holder: Hagood et al.
License: This is an open access article distributed under the terms of the Creative Commons Attribution License, which permits unrestricted use, distribution, reproduction and adaptation in any medium and for any purpose provided that it is properly attributed. For attribution, the original author(s), title, publication source (PeerJ) and either DOI or URL of the article must be cited.
License URL: https://creativecommons.org/licenses/by/4.0/

Keywords: Urology, Guidelines, Adherence

Funding: The authors received no funding for this work.

==============================
Introduction

Over the years, funding for urologic diseases has witnessed a steady rise, reaching $587 million in 2020 from $541 million in 2018. In parallel, there has been a notable increase in the total number of urology journals from 2011 to 2018. This surge in research funding and journal publications calls for urologists to effectively navigate through a vast body of evidence to make the best evidence-based clinical decisions. Our primary objective was to assess the “instructions for authors” of these journals to determine the extent of endorsement of reporting guidelines for common study designs in medical research.

Methods

Top urology journals were identified using the 2021 Scopus CiteScore and confirmed via Google Scholar Metrics h5-index. In a masked, duplicate manner, two investigators retrieved data from the “instructions for authors” webpages of the included journals. For each journal investigated in our study, the following data were extracted: journal title, 5-year impact factor, email responses of journal editors, mention of the EQUATOR Network in the “instructions for authors,” mention of the ICMJE in the “instruction for authors,” geographical region of publication and statements about clinical trial registration.

Results

Of the 92 urology journals examined, only one-third (32/92) mentioned the EQUATOR network in their “instructions for authors.” A total of 17 journals (17/92, 18.5%) did not mention a single reporting guideline. The most endorsed guideline was CONSORT at 67.4% (62/92). Clinical trial registration was not mentioned by 28 (30%), recommended by 27 (29%), and required by 37 journals (40%).

Conclusion

Our findings indicate that urology journals inconsistently endorse reporting guidelines and clinical trial registration. Based on these results, we propose that urology journals adopt a standardized approach, incorporating explicit requirements for reporting guidelines such as those listed on the EQUATOR Network and clinical trial registration for all relevant study designs. Specifically, journals should consider clearly stating mandatory or recommended guidelines for clinical trials, observational studies, and systematic reviews, among others. Future efforts should focus on evaluating the implementation of these policies and identifying barriers that hinder their adoption.

Introduction

Over the years, funding for urologic diseases in the United States has witnessed a steady rise, reaching $587 million in 2020 from $541 million USD in 2018 (American Urological Association, 2024). In parallel, there has been a notable increase in the total number of urology journals from 2011 to 2018 (AlRyalat et al., 2021). This surge in research funding and journal publications calls for urologists to effectively navigate through a vast body of evidence to make the best evidence-based clinical decisions. However, optimal research production is not attained if its individual components are not sufficiently reported (Butcher et al., 2022). Therefore, journals are encouraged to advertise reporting guidelines to encourage high quality reporting, enabling urologists to critically appraise the research evidence based on transparently reported methodological decisions.

Although reporting guidelines can serve as checklists for authors to improve research quality, foster transparency, and mitigate poor methodological conduct that may lead to bias, they encompass much more by providing important conceptual explanations that enhance understanding of proper reporting (Turner et al., 2012; Chan et al., 2014). The Enhancing the QUAlity of Transparency Of health Research (EQUATOR) Network is an international collaboration that is dedicated to the progress of reporting health research studies (EQUATOR Network, 2024). The EQUATOR Network has compiled over 500 reporting guidelines for various research designs. Among the most frequently used guidelines this library hosts are the Consolidated Standards of Reporting Trials (CONSORT) guideline for randomized controlled trials, the Preferred reporting Items for Systematic Reviews (PRISMA) guideline for systematic reviews (Page et al., 2021), and the Strengthening the Reporting of Observational Studies in Epidemiology (STROBE) guidelines for observational studies (Von Elm et al., 2007). Using reporting guidelines to guide research practices helps facilitate the generation of evidence-based research, making it particularly valuable in the context of clinical trials.

Clinical trial registration offers another effective approach to mitigating reporting bias in the medical literature (Won et al., 2019). Lindsley et al. (2022) analyzed systematic reviews of clinical trials from 2015 to 2019 and found that clinical trial registration was linked with a low risk of bias for five of the six evaluated domains. Given the benefits of clinical trial registration, the International Committee of Medical Journal Editors (ICMJE) mandates that all journals published under its umbrella will consider clinical trials for publication only if they were registered before the first patient was enrolled (ICMJE, 2024a). While non-ICMJE member journals that mention the committee in their instructions for authors page are encouraged to follow its guidelines, it is not a requirement.

To date, only one study in urology has assessed journal publication policies concerning reporting guideline recommendations. In the 2012 study, Kunath et al. (2012) examined 55 urology journals to assess the endorsement of 11 reporting guidelines in the “instructions for authors” of these journals. Building upon the insights from this study, our study aims to expand the scope by including a broader range of reporting guidelines and also examining clinical trial registration policies of the top urology journals. We also assessed journal policies regarding clinical trial registration.

Methods

Study design

We conducted a cross-sectional study of urological journal requirements regarding reporting guidelines and clinical trial registration policies according to the STROBE guidelines (STROBE, 2024). Data was obtained using journal websites and their respective “instructions for authors” webpages. This study is based on a protocol designed a priori. The institutional review board deemed that no human subjects were used in this study. The protocol, raw data, extraction forms, and standardized email prompt have been uploaded on Open Science Framework to foster transparency and reproducibility (OSF, 2024).

Search strategy

Investigators, aided by the medical research librarian, identified journals that met the eligibility criteria. Using the 2021 Scopus CiteScore tool we were able to identify a list of top urology journals with high CiteScores (Baas et al., 2020). To ensure the accuracy of the identified journals found by Scopus, we also used Google Scholar Metrics h5-index to affirm the top-twenty urology journals (Vine, 2006). Non-English language journal websites were included by using Google Translate to translate the “instructions for authors” webpage to English. Journals were excluded from our study if they were discontinued, or if they did not publish clinical urology research.

Data collection process

In a masked, duplicate manner, two investigators retrieved data from the “instructions for authors” webpages of the included journals, checking these pages from approximately December 22, 2022, to January 7, 2023. This was achieved using a standardized Google Form that investigators created a priori. After data extraction was completed, investigators were unmasked to one another’s data to reconcile responses. A third investigator was available to resolve disputes should they have persisted after reconciliation.

Data items

For each journal investigated in our study, the following data were extracted: journal title, 5-year impact factor, email responses of journal editors, mention of the EQUATOR Network in the “instructions for authors,” mention of the ICMJE in the “instruction for authors,” and geographical region of publication (i.e., North America, Europe, Asia, or other). We extracted statements about the reporting guideline requirements from each journal’s “instructions for authors.” The full list of reporting guidelines assessed and their respective study designs can be found on Table 1. For each journal, we also extracted statements about clinical trial registration in a similar fashion.

Table 1 Reporting guidelines and study designs.

Study design	Respective reporting guideline	
Animal research	ARRIVE	
Case reports	CARE	
Clinical trials	CONSORT	
Clinical trial protocols	SPIRIT	
Diagnostic accuracy	STARD	
TRIPOD	
Economic evaluations	CHEERS	
Observational studies in epidemiology	MOOSE	
STROBE	
Qualitative research	COREQ	
SRQR	
Quality improvement	SQUIRE	
Systematic reviews and meta-analyses	PRISMA	
QUOROM	
Systematic review and meta-analysis protocols	PRISMA-P	
Note:

Abbreviations: ARRIVE, Animal Research: Reporting of In Vivo Experiments; CARE, Case Reports guidelines checklist; CHEERS, Consolidated Health Economic Evaluation Reporting Standards; CONSORT, Consolidated Standards of Reporting Trials; COREQ, Consolidated Criteria for Reporting Qualitative Research; MOOSE, Meta-Analysis of Observational Studies in Epidemiology; PRISMA, Preferred Reporting Items for Systematic Reviews and Meta-Analyses; PRISMA-P, Preferred Reporting Items for Systematic Review and Meta-Analysis Protocols; QUORUM, Quality of Reporting of Meta-analyses; SPIRIT, Standard Protocol Items: Recommendations for Interventional Trials; SQUIRE, Standards for Quality Improvement Reporting Excellence; SRQR, Standards for Reporting Qualitative Research; STARD, Standards for Reporting Diagnostic Accuracy Studies; STROBE, Strengthening the Reporting of Observational Studies in Epidemiology; TRIPOD, Transparent Reporting of a Multivariate Prediction Model for Individual Prognosis or Diagnosis.

For each data point, it was determined if a given reporting guideline or study registry was recommended, required, not required, or not mentioned. Terms/phrases in the “instructions for authors” such as “required,” “must,” “need,” “mandatory,” and “studies will not be considered for publication unless…” were interpreted as “required.” Terms such as “recommended,” “encouraged,” “should,” and “preferred” were interpreted as “recommended.” Additionally, phrases like “can,” “may if they wish,” and “might” were classified as “not required,” indicating that while authors have the option to follow these guidelines, they are not obligatory. If terms/phrases were unclear, a decision was agreed upon by two study investigators with a third available to adjudicate if needed.

The editorial office of each journal was contacted via email to prevent an incorrect assessment of the reporting guidelines on study designs that journals do not accept. We inquired about the acceptance of the following study designs: clinical trials,systematic reviews, observational studies, meta-analyses, diagnostic accuracy studies, in-vivo experiments, case reports, economic evaluations, qualitative research, quality improvement, clinical trial protocols, and diagnostic and prognostic studies. These study designs and their corresponding reporting guidelines are depicted in Table 1. For each journal, emails were sent to the Editors-in-Chief once a week for three consecutive weeks to optimize response rates (Hoddinott & Bass, 1986). Investigators used a standardized email prompt to minimize variability in communication efforts. If there was no email response after 3 weeks, the journal was assessed on all aforementioned data items.

Outcomes

The primary outcome of our investigation was the proportion of urology journals’ “instructions for authors” that require/recommend the use of reporting guidelines for common study designs. The secondary outcome was the proportion of urology journals that require/recommend the registration of clinical trials.

Data synthesis

We summarized data using descriptive statistics using R (version 4.2.1; R Core Team, 2022) and Rstudio (R Studio Team, 2022) which is a valuable tool that allows for conducting the advanced statistical analyses and data visualization, which are integral to our research methodology. Descriptive statistics included: (1) the proportion of journals requiring/recommending each guideline extracted and (2) the proportion of journals requiring/recommending clinical trial registration. Bias analysis was unnecessary and was not performed in our study as this was a direct assessment of journal web pages rather than at the individual study level.

Results

Journal characteristics

Our initial search identified 99 urology journals for screening. We ultimately extracted data from 92 journals in our final sample (Fig. 1). Journals that did not accept specific study designs were excluded from our analysis of the total 92 journals.

Figure 1 Flow diagram of journal selection.

Impact factor and geographical region

Journals from North America exhibit the highest median impact factor, followed by Europe and Asia. Europe demonstrates the greatest variability in impact factors, while North America shows a more concentrated distribution. Asian journals have the lowest range of impact factors, with fewer outliers. These regional variations in impact factor suggest differences in the influence and visibility of urology research across geographic regions, which may reflect disparities in research output and dissemination. This graph can be viewed in File S1.

Reporting guidelines

Of the 92 urology journals examined, only one-third (32/92) mentioned the EQUATOR network in their “instructions for authors.” A total of 17 journals (17/92, 19%) did not mention a single reporting guideline. Out of the evaluated guidelines, clinical trial registration was the most endorsed entity, with 70% (64/92) of the journals either recommending or requiring its implementation for publication. The most supported guideline was CONSORT at 67% (62/92), followed by PRISMA which was recommended or required by 61% (56/92). The least endorsed guideline was tied by QUOROM and MOOSE with 15% (14/92) of journals recommending or requiring its implementation, followed by SQUIRE at 25% (23/92), and SRQR at 26% (24/92). PRISMA-P was not mentioned in 70% (64/92) of the journals, underscoring the significant gap in awareness and adoption of this guideline. This highlights an important area for improvement in journal practices regarding reporting standards. A complete description of the included journals and their policies are reflected in Table 2.

Table 2 Journal guidelines.

Name of journal	5 year impact factor	Region	Mentions Equator Network	Mentions ICMJE	CONSORT	MOOSE	QUOROM	PRISMA	STARD	STROBE	ARRIVE	CARE	CHEERS	SRQR	SQUIRE	SPIRIT	COREQ	TRIPOD	PRISMA-P	Clinical Trial Registration	
European Urology	21.754	Europe	No	Yes	✹	–	–	✹	✹	✹	✹	–	–	–	–	–	–	–	–	✓	
The Journal of Urology	6.881	North America	Yes	Yes	✓	–	✹	✹	✹	✹	✹	✹	✹	✹	✹	✹	✹	✓	✹	✓	
European Urology Focus	5.747	Europe	No	No	–	–	–	✹	–	–	–	–	–	–	–	–	–	–	–	–	
Prostate Cancer and Prostatic Diseases	5.469	Europe	No	Yes	✓	–	–	✹	–	–	✹		–	–	–	–	–	–		✓	
BJU International	5.417	Europe	Yes	Yes	✹	✹	–	✹	–	–	–	–	–	–	–	✹	–	–		✓	
LGBT Health	5.15	North America	Yes	Yes	✹	–	✹	✹	✹	✹	✹	✹	✹	✹	✹	✹	✹	✹	✹	✹	
Journal of Sexual Medicine	4.883	Europe	No	Yes	✓	–	✓	✓	✓	✓	✹			–	✓		✓	✓		✹	
Sexual Medicine Reviews	4.836	Europe	No	Yes	✓	✓	✓	✓	✹	✓	✹			–	✓		✓	–		✓	
World Journal of Men’s Health	4.503	Asia	No	Yes	✹	–	–	✓	✹	✹	–	–	–	–	–	–	–	–	–	✓	
American Journal of Physiology-Renal Physiology	4.17	North America	No	Yes	–	–	–	✓	–	–	✹	–	–	–	–	–	–	–	–	–	
Andrology	3.842	Europe	No	No	–	–	–	–	–	–	–	–	–	–	–	–	–	–	–	✓	
Prostate	3.817	North America	Yes	Yes	✹	–	✹	✹	✹	✹	✹	✹	✹	✹	✹	✹	✹	✹	✹	✓	
World Journal of Urology	3.737	Europe	Yes	Yes	✹	–	–	✹	✹	✹	✹	✹	✹	✹	✹	✹	✹	✹	✹	✓	
Asian Journal of Andrology	3.678	Asia	Yes	Yes	✹	–	✹	✹	✹	✹	✹	✹	✹	✹	✹	✹	✹	✹	✹	✓	
Advances in Urology	3.676	North America	No	No	✹	–	–	✹	✹	✹	✹	✹	–	✹	–	–	–	–	–	✹	
Kidney Research and Clinical Practice	3.619	Asia	No	Yes	–	–	–	–	–	–	–	–	–	–	–	✹	–	–	–	✹	
Urologic Oncology: Seminars and Original Investigations	3.498	North America	No	Yes	✹	–	–	–	–	–	✹		–	–	–		–	–	–	–	
Minerva Urology and Nephrology	3.42	Europe	No	Yes	✹	–	–	✹	–	–	–	–	–	–	–	–	–	–	–	–	
Therapeutic Advances in Urology	3.182	Europe	Yes	Yes	✓	✹	–	✓	✹	✹		✹	✹	–	–	✹	–	–	–	✓	
CardioRenal Medicine	3.176	Europe	Yes	Yes	✓	✹	–	✹	✹	✹	✓	✹	✹	✹	✹	✹	✹	✹	✹	✹	
Nephron	3.081	Europe	Yes	Yes	✓	–	–	–	–	–	✓	✹	–	–	–	✓	–	–	✓	✓	
Bladder Cancer	3.061	Europe	Yes	Yes	✹	✹	–	✹	✹	✹	✹	✹	✹	✹	✹	✹	✹	✹	✹	–	
Systems Biology in Reproductive Medicine	3.061	Europe	No	Yes	–	–	–	–	–	–	–	–	–	–	–	–	–	–	–	✓	
Abdominal Radiology	3.039	North America	Yes	Yes	✹	✹	–	✹	✹	✹	✹	✹	✹	✹	✹	✹	✹	✹	✹	✓	
Translational Andrology and Urology	3.032	Asia	Yes	Yes	✹	–	–	✹	✹	✹	✹	✹	–	–	–	✹	–	✹	–	✹	
International Journal of Urology	3.011	Australia	No	Yes	✹	–	–	–	–	–	–	–	–	–	–	–	–	–	–	✹	
Current Urology Reports	3.002	North America	No	Yes	–	–	–	–	–	–	–	–	–	–	–	–	–	–	–	✹	
European Urology Open Science	3	Europe	Yes	Yes	✹	✹	–	✹	–	–	✹	–	–	–	–	–	–	–	–	✓	
Urolithiasis	2.959	Europe	Yes	Yes	✹	✹	✹	✹	✹	✹	✹	✹	✹	✹	✹	✹	✹	✹	✹	✓	
Clinical Genitourinary Cancer	2.817	North America	No	Yes	✹	–	–	–	–	–	✹	–	–	–	–	–	–	–	–	✹	
Asian Journal of Urology	2.784	Asia	No	Yes	✓	–	–	✹	–	–	✹	–	–	–	–	–	–	–	–	✹	
Andrologia	2.775	Europe	No	No	✹	–	–	✹	✹	✹	✹	✹	–	✹	–		–	–	–	✹	
Journal of Endourology	2.76	North America	Yes	Yes	✹	–	–	✹	✹	✹	✹	✹	✹	✹	✹	✹	✹	✹	✹	–	
Prostate International	2.702	Asia	No	Yes	✹	–	–	–	–	–	✹	–	–	–	–	–	–	–	–	✹	
Urologic Clinics of North America	2.685	North America	No	Yes	–	–	–	–	–	–	–	–	–	–	–	–	–	–	–	–	
International Neurourology Journal	2.658	Asia	No	Yes	–	–	–	–	–	–	–	–	–	–	–	–	–	–	–	✹	
Arab Journal of Urology	2.646	North America	Yes	Yes	✹	–	–	✓	–	–	–	–	–	–	–	–	–	–	–	✓	
Current Opinion in Urology	2.63	North America	No	Yes	–	–	–	–	–	–	–	–	–	–	–	–	–	–	–	–	
Journal of Endourology Case Reports	2.619	North America	Yes	Yes	–	–	–	–	–	–	–	–	–	–	–	–	–	–	–	–	
Sexual Medicine	2.602	North America	No	Yes	✓	–	✓	✓	✓	✓	✓	✓	–	–	✓	–	✓	✓	–	✹	
Neurourology and Urodynamics	2.533	North America	No	Yes	✓	–	–	✓	–	–	–	–	–	–	–	–	–	–	–	✓	
International Urogynecology Journal	2.426	Europe	No	No	✹	–	–	–	–	–	–		–	–	–	–	–	–	–	✹	
Investigative and Clinical Urology	2.41	Asia	No	Yes	–	–	–	✓	–	–	–	–	–	–	–	–	–	–	–	✹	
Basic and Clinical Andrology	2.323	Europe	Yes	Yes	✓	–	–	✹	✹	✹	✹	✹	✹	–	–	✹	✹	✹	✹	✓	
BMC Urology	2.317	Europe	Yes	Yes	✓	–	–	✹	✹	✹	✹	✹	✹	–	–	✹	✹	✹	✹	✓	
International Urology and Nephrology	2.302	Europe	Yes	Yes	✹	✹	–	✹	✹	✹	✹	✹	✹	✹	✹	✹	✹	✹	✹	✓	
International Journal of Impotence Research	2.276	Europe	No	Yes	✓	–	–	✹	–	–	✹	–	–	–	–	–	–	–	–	✓	
Journal of Pediatric Urology	2.102	Europe	No	Yes	✹	–	–	–	–	✹	✹	–	–	–	–	–	–	–	–	✓	
Urologia Internationalis	2.075	Europe	Yes	Yes	✓	–	–	✹	✹	✹	✓	✹	✹	✹	✹	✓	✹	✹	✓	✹	
Scandinavian Journal of Urology	1.911	Europe	No	Yes	–	–	–	–	–	–	–	–	–	–	–	–	–	–	–	✓	
Current Urology	1.82	North America	Yes	Yes	✹	–	–	✹	–	✹	–	✹	–	–	–	–	✹	–	–	✓	
Prostate Cancer	1.81	Other	No	No	✹	–	–	✹	✹	✹	✹	✹	–	✹	–	–	–	–	–	✹	
Wideochirurgia I Inne Techniki Maloinwazyjne	1.65	Europe	No	Yes	–	–	–	–	–	–	–	–	–	–	–	–	–	–	–	–	
Research and Reports in Urology	1.614	Europe	No	Yes	–	–	–	–	–	–	–	–	–	–	–	–	–	–	–	✓	
Journal of Lasers in Medical Sciences	1.61	Other	Yes	Yes	✹	–	–	✹	✹	✹	✹	✹	✹	✹	✹	✹	✹	✹	✹	✓	
International braz j urol : official journal of the Brazilian Society of Urology	1.57	Other	No	Yes	–	–	–	✓	–	–	–	–	–	–	–	–	–	–	–	✓	
Urology	1.51	North America	No	No	✓	–	–	✹	–	–		–	–	–	–		–	–		–	
Urology Journal	1.508	Asia	No	No	✹	–	–	✹	–	–	–	–	–	–	–	–	–	–	–	✓	
Female Pelvic Medicine and Reconstructive Surgery	1.45	North America	Yes	Yes	✹	✹	–	✹	✹	✹	✹	✹	✹	✹	✹	✹	–	✹	–	✹	
LUTS: Lower Urinary Tract Symptoms	1.374	Other	No	Yes	✹	–	–	–	–	–	–	–	–	–	–	–	–	–	–	✹	
Central European Journal of Urology	1.17	Europe	No	No	–	–	–	–	–	–	–	–	–	–	–	–	–	–	–	–	
Renal Replacement Therapy	1.17	Europe	No	Yes	✓	–	–	✹	✹	✹	✹	✹	✹	–	–	✹	✹	✹	✹	✓	
Journal of Renal Injury Prevention	1.13	Asia	Yes	Yes	✹	–	✹	✹	✹	✹	✹	✹	✹	✹	✹	✹	✹	✹	✹	–	
Turkish Journal of Urology	1.053	Asia	Yes	Yes	✓	–	–	✓	✓	✓	✓	–	–	✓	–	✓	–	–	✓	✓	
International Journal of Urological Nursing	0.91	Europe	No	Yes	✹	–	–	–	–	–	–	–	–	–	–	–	–	–	–	✹	
Urology Annals	0.901	Asia	No	Yes	✹	✹	✹	–	✹	✹	–	–	–	–	–	–	–	–	–	✓	
Progres en Urologie	0.822	Europe	No	No	✹	–	–	✹	–	–	–	–	–	–	–	–	–	–	–	–	
Actas Urologicas Espanolas	0.72	Europe	No	No	✓	–	–	✓	–	✓	✹	–	–	–	–	–	–	–	–	✓	
Indian Journal of Urology	0.638	Asia	No	Yes	–	–	–	–	–	–	–	–	–	–	–	–	–	–	–	–	
Journal of Men’s Health	0.537	Asia	No	Yes	–	✹	–	✹	–	–	–	–	–	–	–	–	–	–	–	–	
Revista Internacional de Andrologia	0.51	Europe	No	No	✹	–	–	✹	✹	✹	✹	✹	–	✹	–	–	–	–	–	✹	
Urology Case Reports	0.489	North America	No	No	–	–	–	–	–	–	–	–	–	–	–	–	–	–	–	–	
Urology Practice	0.47	North America	No	Yes	✓	–	–	✓	–	–	–	–	–	–	–	–	–	–	–	✓	
Aktuelle Urologie	0.423	Europe	Yes	Yes	✹	–	✹	✹	✹	✹	✹	✹	✹	✹	✹	✹	✹	✹	✹	✹	
African Journal of Urology	0.422	Europe	Yes	Yes	✹	–	✹	✹	✹	✹	✹	✹	✹	✹	✹	✹	✹	✹	✹	–	
Nephro-Urology Monthly	0.386	Europe	Yes	Yes	✹	✹	✹	✹	✹	✹	✹	✹	✹	✹	✹	✹	✹	✹	✹	✓	
Der Urologe	0.33	Europe	No	No	–	–	–	–	–	–	–	–	–	–	–	–	–	–	–	–	
IJU Case Reports	0.33	Other	No	Yes	–	–	–	–	–	–	–	–	–	–	–	–	–	–	–	✹	
Journal of Clinical Urology	0.312	Europe	Yes	Yes	✹	–	✹	✹	✹	✹	✹	✹	✹	✹	✹	✹	✹	✹	✹	✓	
Urological Science	0.28	Asia	No	Yes	✹	–	–	✹	✹	✹	✹	✹	–	–	✹	–	–	–	–	✓	
Onkourologiya	0.24	Asia	No	Yes	–	–	–	–	–	–	–	–	–	–	–	–	–	–	–	–	
Tijdschrift voor Urologie	0.18	Europe	No	Yes	–	–	–	–	–	–	–	–	–	–	–	–	–	–	–	–	
Urologia Colombiana	0.155	Other	Yes	Yes	✹	–	–	✹	✹	✹	–	✹	–	–	–	–	–	–	–	✹	
Urogynaecologia International Journal	0.13	Europe	No	No	✹	–	–	–	–	–	–	–	–	–	–	–	–	–	–	✹	
Open Urology and Nephrology Journal	0.103	Europe	Yes	Yes	✹	✓	–	✓	✓	✓	✹	✓	✓	–	–	–	✓	✓	–	✹	
Indonesian Journal of Obstetrics and Gynecology	0.1	Asia	No	No	–	–	–	–	–	–	–	–	–	–	–	–	–	–	–	–	
Japanese Journal of Urology	0.083	Asia	No	No	–	–	–	–	–	–	–	–	–	–	–	–	–	–	–	–	
Revista mexicana de urologia	0.06	Other	No	Yes	–	–	–	–	–	–	–	–	–	–	–	–	–	–	–	–	
Progres en Urologie-FMC	0.02	Europe	No	No	–	–	–	–	–	–	–	–	–	–	–	–	–	–	–	–	
Journal fur Urologie und Urogynakologie	0.017	Europe	No	No	–	–	–	–	–	–	–	–	–	–	–	–	–	–	–	–	
Seksuologia Polska (Journal of Sexual and Mental Health)	NA	Europe	No	Yes	–	–	–	–	–	–	–	–	–	–	–	–	–	–	–	–	
Ukrainian Journal of Nephrology and Dialysis	NA	Asia	No	Yes	–	–	–	–	–	–	–	–	–	–	–	–	–	–	–	–	
Note:

✓ = required, ✹ = recommended, – = not mentioned, boxes filled in with grey = study does not accept.

Clinical trial registration

Of our sample, 79% (73/92) journals mentioned ICMJE in their respective “instructions for authors.” Clinical trial registration was not mentioned by 28 (30%), recommended by 27 (29%), and required by 37 journals (40%).

Discussion

Our findings demonstrate a notable deficiency in the extent to which urology journals mention reporting guidelines and clinical trial registration policies in their “instructions for authors” webpages. Nearly one-fifth of our sample failed to refer to a single reporting guideline and approximately one-third of our sample failed to provide a statement regarding clinical trial registration policies. Further, we found that fourteen journals referenced QUOROM, which has largely been superseded by PRISMA, giving concern that some “instructions for authors” may be outdated (Moher et al., 2009). In 2012, a similar study was conducted by Kunath et al. (2012) finding that urology journals had suboptimal endorsement of reporting guidelines. However, our findings showed a marginal improvement in guideline reporting usage in comparison over a decade (Kunath et al., 2012). Additionally, in the previous study, only one journal mentioned the EQUATOR Network whereas ours acknowledged it in over a quarter of the included journals. Given our results, it is evident that there has been some advancement in the endorsement and awareness of guideline tools over the last decade. This may be attributable to the EQUATOR Network’s efforts towards improving familiarity with reporting guidelines by researchers and journal editors alike (Simera et al., 2010; Altman & Simera, 2016).

While our study demonstrates improved advocacy of reporting guidelines in urology journals recently, it remains uncertain how effectively these guidelines are being embraced and endorsed by journals. This potential disconnect between journal and author expectations for publication could hinder the production of high-quality and reproducible studies. For instance, Koch et al. (2016a, 2016b) reported that the explicit mention of reporting guidelines in urogynecology is relatively low, although there has been some improvement in randomized controlled trials; nonetheless, adherence remains notably insufficient. Similarly, Cook et al. (2018) observed that less than half of dermatology clinical trials published in journals mandating CONSORT adherence included a CONSORT flow diagram, a crucial component of the guideline, alongside their manuscripts. This variability in both journal advocacy and researcher adherence across different medical fields raises concerns about the adequacy of reported research, influencing clinical practice guidelines and clinical decision-making. To address this issue, improving the publication policies of academic journals is crucial. Some authors suggest implementing regular audits to update policies in line with current practices and expectations (De Kleijn & Van Leeuwen, 2018). If adopted, journals could ensure their “instructions for authors” are explicitly guided by evidence-supported reporting guidelines, such as CONSORT and PRISMA. These journals might also mandate prospective authors to submit a thoroughly completed checklist relevant to their study design along with their manuscript for consideration of publication. Such measures would help promote transparent and high-quality reporting of research, enhancing the overall reliability and influence of clinical research in urology and other medical fields.

Many academic journals now require the pre-registration of clinical trials as a prerequisite for publication, aiming to uphold ethical standards in research involving human subjects and ensuring high-quality outcomes (Harvey, 2017). The International Committee of Medical Journal Editors (ICMJE) goes a step further, considering clinical trial registration an ethical obligation for researchers, viewed as the “single most valuable tool we have to ensure unbiased reporting” (ICMJE, 2024b; Wallach & Krumholz, 2019; Weber, Merino & Loder, 2015). Despite this strong endorsement by the ICMJE, our study revealed a concerning gap in clinical trial registration policies among urology journals. While approximately three-quarters of the sampled journals mention the ICMJE in their “instructions for authors,” less than half of them actually require clinical trial registration. Additionally, only a quarter of the journals merely recommend this practice. It appears, however, this deficiency is not unique to urology. For example, Hooft et al. (2014) found in a survey of 757 ICMJE supporting journals, approximately half of their sample explicitly advocated for prospective clinical trial registration in their “instructions for authors”. Further, upon contacting the editorial teams of these journals, they uncovered that only 18% actually cross-checked manuscript submissions against public registries and that nearly 70% of journals would accept retrospectively registered trials (Hooft et al., 2014). Such practices raise concerns about potential selective reporting biases that could influence research outcomes (Zhou et al., 2022).

The editorial team of a journal plays a pivotal role as the gatekeeper of well-designed and published research, holding the power to influence how authors conduct and report their findings. As such, it is essential for journals to provide clear and explicit instructions for authors and establish easy channels for communication to address any ambiguities. In our study, we found journal homepages to be frequently difficult to navigate and composed of vague language which made data extraction a challenging process. To add to this, less than half of our sample followed up to our email correspondence by the time of this study’s conclusion. Not only does this pose a threat to our study’s accuracy regarding journal publication policies, but also interferes with the publication process for prospective authors who may misinterpret a journal’s requirements and recommendations, effectively reducing transparency and reproducibility of published works (Rauh et al., 2022). In 2020 an article was published describing four ways journals can aid reproducibility of research by using structure, transparency, accessibility, and reporting (S.T.A.R.) as a method which allows journals to enforce and introduce important policies for authors and editors (Elsevier, 2021). We recommend, therefore, that journals take action to improve the clarity of their webpages and offer reliable methods of contact to reduce strain between expectations and fulfilled requirements by authors.

Strengths and Limitations

This study has notable strengths. First, rigorous data screening and extraction processes were completed in a masked, duplicate fashion which is considered the “gold standard” for cross-sectional analyses and maximizes the accuracy of the data collection process with mitigation of errors. Second, this study is based on a protocol designed a priori. Similarly, this study itself was conducted according to the STROBE guidelines for cross-sectional analyses. Both these efforts were made to improve the transparency and reproducibility of our work for future research to be conducted. A limitation of our study was the finite email response from editorial teams, leaving us uncertain about the completeness of our results. While we ensured “instructions for authors” were interpreted with care, it is possible these instructions do not fully reflect journal policies. However, it should be considered that we undertook this project with the same context and information prospective authors would have when attempting to publish their works, demonstrating that regardless of this study’s results, improvements in journal clarity and communication are critical for the field of urology.

Conclusion

Our investigation reveals that urology journals lack consistent endorsement of reporting guidelines and clinical trial registration requirements. To promote the quality of research reporting, foster transparency, and prevent reporting bias, it is imperative for urology journals to adopt well studied and validated reporting guidelines and enforce prospective clinical trial registration. Therefore, we recommend that urology journals consider implementing methods to improve their publication policies. Future research may be directed at developing strategies to accomplish this.

Supplemental Information

Supplemental Information 1 Regional Variability in Impact Factors of Urology Journal.

Box plot showing the distribution of urology journal impact factors by region: North America, Europe, and Asia.

Supplemental Information 2 STROBE Checklist.

Additional Information and Declarations

Competing Interests

Author Contributions

Data Availability

Matt Vassar received funding from the National Institute on Drug Abuse, the National Institute on Alcohol Abuse and Alcoholism, the U.S. Office of Research Integrity, Oklahoma Center for Advancement of Science and Technology, and internal grants from Oklahoma State University Center for Health Sciences-all outside of the present work.

Alex Hagood performed the experiments, analyzed the data, prepared figures and/or tables, and approved the final draft.

Joseph Case performed the experiments, analyzed the data, prepared figures and/or tables, and approved the final draft.

Trevor Magee performed the experiments, analyzed the data, prepared figures and/or tables, and approved the final draft.

Caleb Smith conceived and designed the experiments, authored or reviewed drafts of the article, and approved the final draft.

Danya Nees conceived and designed the experiments, authored or reviewed drafts of the article, and approved the final draft.

Griffin Hughes conceived and designed the experiments, analyzed the data, authored or reviewed drafts of the article, and approved the final draft.

Matt Vassar conceived and designed the experiments, authored or reviewed drafts of the article, and approved the final draft.

The following information was supplied regarding data availability:

This is a systematic review/meta-analysis.

The data is available at Open Science Framework: Smith, C. A., Vassar, M., Hughes, G., Nees, D., Magana, K., Duncan, J., … Gardner, T. (2023) Endorsement of reporting guidelines and clinical trial registration by clinical journals. Available at osf.io/wrtke.

The data is available at Zenodo: Lanshakov, D., Shaburova, E., Sukhareva, E., Bulygina, V., Drozd, U., Larionova, L., Gerashchenko, T., Shnaider, T., Denisov, E., & Kalinina, T. (2024). PeerJ Supplemet 2024 [Data set]. Zenodo. https://doi.org/10.5281/zenodo.13773297.

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
