# Peer review of "Endorsement of reporting guidelines and clinical trial registration across urological medical journals: a cross-sectional study"

_PeerJ, doi:10.7717/peerj.18619_

## Round 0.1 · original submission · Major Revisions

· Academic Editor

Major Revisions

My apologies for the extremely slow review. This proved to be a particularly difficult manuscript for which to find suitable reviewers without conflicts of interest. I have supplemented our reviewers’ comments with my own below. Reviewer #1 has made some very useful comments, focusing on improving the clarity of your work, but also inviting you to consider doing more with your data. These are valuable suggestions, and each warrants a point-by-point response with revisions made in response to each tracked in the updated manuscript. Reviewer #2 has suggested two additional references for you to consider. Please note that you should only add these if you find them useful and you are entirely free to add alternative references that serve a similar purpose.

Overall, the writing was relatively error-free, but the text doesn’t always ‘flow’ well for me. I appreciate that this is a rather vague and unhelpful comment, but thinking about this when revising the work would help readability.

It’s unclear why impact factor was collected, but this does seem a useful variable to stratify the data—see Reviewer #1’s comments including their suggested version of Table 2 (their final comment). You could do this by IF Quartile, for example, as well as overall. If there are other ways to think about the journals, and I’m not suggesting further data collection unless you believe it could help (location of journal, age of journal, whether the journal has a print edition or only electronic, etc.), it would be useful to also ask if breaking down the responses by these might help with the story. I’m not suggesting this as a gratuitous extension of your work, but something to evaluate whether it could help. For example, if higher impact journals are more likely to mention and require the outcomes here, that makes a useful point for me and could encourage journals to make aspirational changes in their instructions/policies.

Specific comments:

Line 50: The reader will wonder why 2011–2018 here?

Line 51: “This surge” doesn’t follow from the previous sentence. More journals doesn’t necessarily mean more funding.

Line 70: See also comments from Reviewer #1 but this is the first mention in the abstract of registration requirements.

Lines 69–72: I think you could try to make your conclusions more specific around the methods and strategies that you are suggesting here.

Line 95: Please be specific about currency and location here.

Line 103: While reporting guidelines are generally implemented with a checklist, I think that this understates the intellectual material in these guidelines, particularly alongside their explanations and elaboration companions when these are available.

Lines 108–112: As an illustration of how used these are, you could mention in passing the citations for the associated articles, perhaps focusing on the journals most likely to be read by urology researchers?

Line 154: As per Reviewer #1’s suggestion, this would be a useful place for you to add the time period during which the instructions were checked.

Line 173: I’m guessing that words such as ‘can’, ‘may if they wish’, ‘might’, etc. were used to identify the ‘not required’ classifications (c.f. “not mentioned” which doesn’t require definition)?

Line 195: Not intending to be glib, but RStudio doesn’t seem any more relevant here than using Microsoft Word.

Line 209: As your denominator is just under 100, each individual journal counts for slightly more than 1%. I suggest integer percentages here and throughout.

Lines 213–214: Is there a reason for reversing the direction of reporting (absence rather than presence) for PRISMA?

Table 2: This seems more useful as a supplementary table to me. Adding a summary table as suggested by Reviewer #1 would allow this and you could also consider graphical representations where these help.

Reviewer 1 ·

Basic reporting

Title: Consider modifying it as “a cross-sectional study”. If it is identified as a review, better to comply with PRISMA rather than STROBE.

Abstract
The paper is titled “Endorsement of …and clinical trial registration…”, but it is mismatched with its primary objective, aiming to “assess the instructions for authors of these journals to determine the extent of endorsement of reporting guidelines…”. The objective seems to have nothing to do with the trial registration. Furthermore, no any content about trial registration in the abstract’s background, methods, and results. It’s necessary to add assessment methods and findings about “trial registration” in the abstract.
Line 57: Better to show how to search and identify target journals, and inclusion and exclusion criteria.

Introductions
Line 98-99: “optimal research production is not attained if its individual components are not sufficiently reported”, need supporting references.
Line 126-131: There is repetition here, e.g. “our study aims to… our primary objective was…”.

Experimental design

Methods
Line 145: “a list of top urology journals…” Only top journals? Or filter the list for all urology journals? If the former one, how to define the “top”?

Validity of the findings

Results
There is no any summary finding about the impact factor and geographical region of journals except for Table 2. Why extract them? Maybe consider conducting some statistical analysis, e.g. regression, to explore their correlations.

Better to present when these data items were extracted, e.g. from March 2023 to April 2023, as website content is always changing.

Line 212: The QUOROM was updated to the PRISMA statement in 2009, so as I see it, they both should not be analyzed separately. In addition, PRISMA has many extensions, so why only extract PRISMA-P? Conversely, there are many extensions for CONSORT, including CONSORT for abstracts, why not extract them?

Table 2: better to have statistical results for each item. Not just described in the results.

Reviewer 2 ·

Basic reporting

The study is reported in a clear and concise way. The authors adhered to the STROBE checklist.

Experimental design

Methods are adequate and replicable.

Validity of the findings

Study results are relevant in order to document the yet insufficient requirements of urology journals to foster good scientific practice and reporting.

Additional comments

I recommend to include the following literature references into the discussion (line 238-240), since they already investigated the endorsement of reporting guidelines in urogynecology journals.

Koch M, Riss P, Umek W, Hanzal E (2016) The explicit mentioning of reporting guidelines in urogynecology journals in 2013: A bibliometric study. Neurourology and urodynamics 35 (3):412-416. doi:10.1002/nau.22726


Koch M, Riss P, Umek W, Hanzal E (2016) CONSORT and the internal validity of randomized controlled trials in Female Pelvic Medicine. Neurourology and urodynamics 35 (7):826-830. doi:10.1002/nau.22811

---

## Round 0.2 · accepted · Accept

· Academic Editor

Accept

Thank you for your revisions. As you can see, neither of our reviewers have raised further points and I'm satisfied with your responses and revisions for my queries. I am delighted to accept your manuscript and look forward to seeing it published in due course. Well done.

Reviewer 1 ·

Basic reporting

no comment

Experimental design

no comment

Validity of the findings

no comment

Additional comments

The author team revised the manuscript accordingly. All my concerns were addressed adequately.

Reviewer 2 ·

Basic reporting

the authors adequately implemented my recommendations- I have no further comment

Experimental design

the authors adequately implemented my recommendations- I have no further comment

Validity of the findings

the authors adequately implemented my recommendations- I have no further comment

Additional comments

the authors adequately implemented my recommendations- I have no further comment